# Olfactory screening of Parkinson's Disease patients and healthy subjects in China and Germany: A study of cross-cultural adaptation of the Sniffin' Sticks 12-identification test

Elmar H. Pinkhardt[ID][1]*, Huijing Liu[2], Di Ma[3], Jing Chen[3], Adrian Pachollek[1], Martin S. Kunz[1], Jan Kassubek[1], Albert C. Ludolph[1], Yining Huang[3], Haibo Chen[2], G. Bernhard Landwehrmeyer[1], Zhaoxia Wang[3], Wen Su[2]*

1 Ulm University, Department of Neurology, Ulm, Germany, 2 Neurology Department, Beijing Hospital, National Center of Gerontology, China, Dong Dan, Beijing, China, 3 Department of Neurology, Peking University First Hospital, Beijing, China

* elmar.pinkhardt@uni-ulm.de (EP); suwendy@126.com (WS)

## Abstract

### Background

Olfactory testing is a useful tool in the differential diagnosis of Parkinson's Disease (PD). Although fast and easy to use, the high intercultural variability of odor detection limits the world-wide use of the most common test sets.

### Objective

The aim of this study was to test one of the most commonly used olfactory tests (Sniffin' Sticks 12-identification test) in an adapted version for a Chinese population of healthy subjects and PD patients.

### Methods

For this purpose, cohorts of 39 Chinese and 41 German PD patients as well as 70 Chinese and 100 German healthy subjects have been examined both with the original and the adapted version of the Sniffin' Sticks test, the latter being designed according to the regional culture.

### Results

The adapted Chinese version of the Sniffin' Sticks 12 identification test proved to discriminate Chinese PD patients from controls with a high specificity but relatively low sensitivity. Yet not all odor exchanges would have been necessary as the original odors including liquorice and coffee showed an equally high identification rate in the Chinese and German cohorts.

**Data Availability Statement:** All relevant data are within the manuscript and its Supporting Information files.

**Funding:** All authors were part of the study group that was funded by the UULM-PUHSC-Joint-Center. The funders had no role in study design, data collection and analysis, decision to publish, or preparation of the manuscript.

**Competing interests:** The authors have declared that no competing interests exist.

## Conclusions

The results showed that the newly adapted test could be used as a screening test for PD related olfactory dysfunction in a Chinese population. However further investigation will be necessary to optimize the selection of odors for the Chinese version of the test.

## Introduction

Increasing evidence suggests the presence of a long prodromal phase of Parkinson's Disease (PD), with nonmotor symptoms that may start decades before diagnosis [1]. One of those highly frequent prodromal symptoms is olfactory dysfunction. It typically occurs 4–8 years before diagnosis, although in some cases it can occur as early as 20 years before the first motor signs [2,3]. Subjects with the combination of hyposmia and idiopathic REM Sleep Behavior Disorder (iRBD) are at high risk for conversion to PD or PD dementia of more than 60% over 5 years [4].

Olfactory dysfunction in neurodegeneration is not unique to PD but can be found also in Alzheimer's disease and Huntington's disease [5], yet patients with Multiple System Atrophy (MSA) or Progressive Supranuclear Palsy (PSP) seem to suffer less from olfactory loss [6,7].

Neuropathological markers of PD could be linked with olfactory dysfunction, as Lewy-body pathology was found in the olfactory bulb, tract and anterior olfactory nucleus. Likewise alpha synuclein pathology was found in all five sub-regions of the primary olfactory cortex [8]. Dopaminergic treatment has no effect on odor identification performance in PD patients [9]. Together with PET studies that found no correlation between cerebral dopamine transporter activity and odor identification, these findings suggest that PD related olfactory loss most probably reflects a non-dopaminergic mechanism [10].

In order to screen for hyposmia there is a large variety of tests that examine an individual's ability to identify, detect, discriminate or remember odors [5] but often such tests differ in reliability, are not equated for non-olfactory task demands and use different odorants.

In addition, our biological infrastructure and our environment have a high impact to our odor identification abilities [11]. Even genetic variability within and between populations has to be taken into consideration [12,13]. As a consequence, adaptation of olfactory tests to a given cultural setting is essential.

One of the most commonly used olfactory tests is the Sniffin' Sticks test-battery (Burghart Messtechnik, Wedel, Germany) that is available as a) the Sniffin' Sticks 12-identification test (SIT-12) and b) an olfactory test consisting of three different subtests with altogether 112 odorant sticks, testing threshold, discrimination and identification. Due to practical reasons the SIT-12 is the more widely used clinical olfactory test in everyday testing in otorhinolaryngological and neurological departments in Germany and other European countries [14]. It has been validated for clinical use in several European countries, i.e. in Great Britain, Germany, Turkey, Portugal and Denmark [14–18]. Several variants of the SIT test [19][20][21] and other odor identification tests [22] have been examined in an Asian population.

Yet this is the first study to investigate olfactory function in a North Chinese (Beijing) and German population of healthy subjects and PD patients by hands of a direct comparison of the SIT-12 test and a modified SIT-12 test both in a Chinese and German age matched cohort. The for China modified version of the SIT-12 test (Ch-SIT-12) takes into account cultural components by exchanging 4 odors as well as their according descriptors for any given odorant. The decision which odors were changed and which odors were chosen as replacement is

based on the study by Shu et al. [21] Due to expert opinion, however, it was decided not to replace the odor shoeleather.

We hypothesized to find a significant difference in the distribution of incorrect answers in the Chinese cohort between the original SIT-12 and the adapted version and that not all odorants were correctly identified by >75% in a normosmic northern Chinese population in the original version. Vice versa, we hypothesized that the adapted Chinese version of the SIT-12 test would result in falsely pathological results (lower than 75% correctly identified odors) in a healthy German cohort, consequently leading to false positive test results when used as a clinical marker for PD. For the patient groups we hypothesized that both the Chinese as well as the German PD patients would perform worse in the non-adapted version of the test in comparison to the adapted version.

## Materials and methods

### Participants

The original SIT-12 test as well as the Ch-SIT-12 test were administered to 80 PD patients (41 German and 39 Chinese) and 170 healthy controls (100 German and 70 Chinese) who were subjectively normosmic and showed no known disease associated with olfactory dysfunction.

Subjects with rhinitis, history of sinunasal disease or known hyposmia out of other reasons than PD were excluded from this study. The study was approved by the Ethical Committees of the University of Ulm, Peking University First Hospital, and Beijing Hospital, and was performed in accordance with the ethical standards laid out by the Declaration of Helsinki. All prospectively examined patients and controls gave written informed consent. The PD diagnoses were made by board-certified neurologists specialized in movement disorders according to the new clinical Movement Disorder Society diagnostic criteria for PD [23].

### Cohorts

Additional to the primary analysis of the healthy probands (HP) we divided the cohort of healthy subjects in two groups for each country. One group of 32 Chinese and 42 German subjects that served as age-matched controls for patients, and a much younger group of 38 Chinese and 38 German subjects between 20 to 40 years of age in order to test if the different socio-environmental setting of these younger probands may have an influence to odor recognition in China and Germany. The patient group consisted of 39 Chinese and 41 German patients. For a summary of all demographic data for all groups and subgroups see Table 1.

### Demographic data and clinical scores

In addition to olfactory testing, disease duration was recorded and the disease severity was examined by the Unified Parkinson's Disease Rating Scale Part III (UPDRS-III). UPDRS-III was performed under ongoing treatment in the On state. For patients and controls the following information was recorded: age, gender, cumulative years of education and smoking habits (pack years). For a summary of all clinical data see Table 1.

### Olfactory testing

The original SIT-12 test consists of 12 pens, filled with those odorants as depicted in Fig 1. The test is executed as a multiple-choice test. According to the suggested procedure by Hummel et al [24] the odor sticks are successively placed about 2–3 cm in front of the nose of the subject for a period of about 3 seconds, then the subject choses an item from a specified list of four descriptors for each stick [24].

**Table 1. Demographic and clinical data for all Parkinson's Disease patients (all PD), the Chinese (Chinese PD) and German (German PD) cohorts of PD; healthy probands and its subgroups of young (20–40 years old) healthy subjects and age matched controls for the PD groups.**

| | all PD | Chinese PD | German PD | all probands | Chinese probands | Chinese young probands | Chinese controls | German probands | German young probands | German controls |
|---|---|---|---|---|---|---|---|---|---|---|
| Number | 80 | 39 | 41 | 170 | 70 | 38 | 32 | 100 | 38 | 42 |
| Gender (female/male) | 32/48 | 20/19 | 12/29 | 105/65 | 43/27 | 21/17 | 22/10 | 62/38 | 23/15 | 24/18 |
| Age years (mean ± SD; min/max) | 65.5 ± 9.3; 39.6/88.1 | 64.1 ± 9.2; 39.6/82.1 | 66.8 ± 9.2; 49.7/88.1 | 45.8 ± 16.7; 21.6/84.7 | 43.7 ± 17.4; 21.8/84.7 | 29.5 ± 5.1; 21.8/39.5 | 60.6 ± 10.1; 41.9/84.7 | 47.2 ± 16.2; 21.6/82.7 | 29.9 ± 5.5; 21.7/40.4 | 62.9 ± 8.6; 52.9/82.7 |
| Education (median (interquartile range)) | 12.0 (6.0) | 12.0 (7.0) | 12.0 (3.38) | 15.0 (5.0) | 16.0 (7.0) | 18.0 (4.0) | 12.0 (6.0) | 15 (13;17) | 15.0 (3.5) | 14.0 (4.0) |
| Smoking (x/all) | 13/80 | 5/39 | 8/41 | 30/170 | 2/70 | 1/38 | 1/32 | 28/100 | 7/38 | 14/42 |
| Pack years (median (interquartile range)) | 0.0 (0.0) | 0.0 (0.0) | 0.0 (0.0) | 0.0 (0.0) | 0.0 (0.0) | 0.0 (0.0) | 0.0 (0.0) | 0.0 (2.0) | 0.0 (0.0) | 0.0 (2.75) |
| SIT-12 (median of correct answers (interquartile range)) | 6.0 (4.0) | 7.0 (3.0) | 5.0 (4.5) | 11.0 (1.0) | 10.0 (2.0) | 10.0 (2.0) | 9.0 (3.0) | 11.0 (2.0) | 11.0 (2.0) | 11.0 (1.25) |
| Ch-SIT 12 (median of correct answers (interquartile range)) | 6.5 (4.0) | 7.0 (4.0) | 6.0 (4.5) | 11.0 (2.0) | 11.0 (1.25) | 11.0 (2.0) | 10.0 (1.0) | 11.0 (2.0) | 11.0 (1.0) | 11.0 (2.25) |
| Disease Duration in months (median (interquartile range)) | 42.0 (108.0) | 24.0 (38.0) | 96.0 (143.0) | | | | | | | |
| UPDRS III (median (interquartile range)) | 25.0 (22.0) | 22.0 (18.0) | 27.0 (22.0) | | | | | | | |

For the adapted Chinese version of the SIT-12 (Ch-SIT-12), 4 odorants (cinnamon, liquorice, coffee and cloves) were replaced with sesame oil, soy sauce, chocolate and garlic. Both patients (PD) and HP in China and Germany performed both versions of the test. The decision to change the above-mentioned odors was based on the familiarity ratings of odors as described by Shu et al [21] and the experience of the Chinese authors of this study. In order to avoid retest effects, the test was performed with all 16 different odors successively presented as one test. The subjects were not informed about the exact purpose of the modifications made to the original SIT-12 test in order to avoid clichés in odor recognition answers. The sequence of the sticks for the German cohort was the original SIT-12 test followed by the 4 new odorants for the Chinese version (Fig 1). For the Chinese cohort the regionalized Chinese odorants replaced the original sticks within the sequence. The sorted-out sticks were annexed. When used as descriptors in other odorants of the original SIT-12 test, the exchanged odors names were used as descriptors in the Chinese version instead. However, a double testing of original and changed descriptors in both groups was not performed. According to the original validation of the SIT-12, we aimed at a successful identification rate of >75% in a normosmic population [14,24].

## Statistical analysis

All data are displayed as median value and interquartile range in brackets. Due to the non-normal distribution of the data, non-parametric tests were used. The Chinese and German

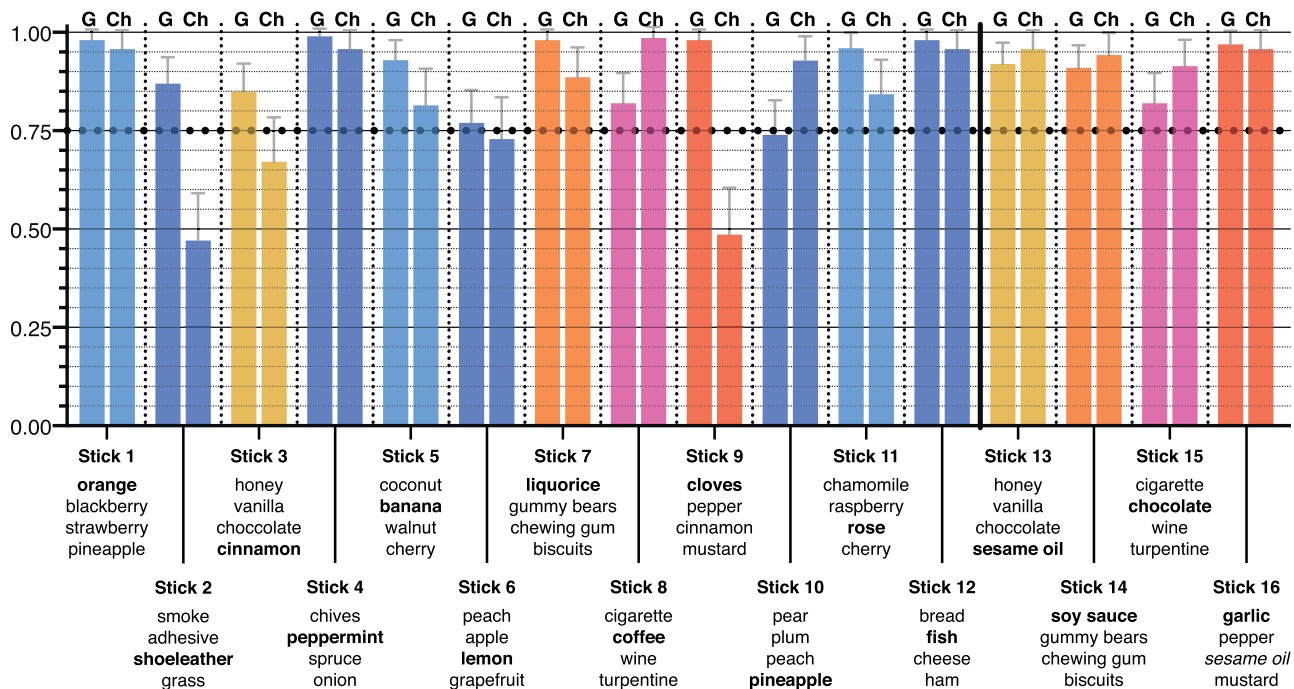

**Fig 1. Odors of all 16 Sniffin' Sticks (bold) and descriptors (changed descriptor for the Chinese version in oblique).** Blue colored sticks are those that remained unchanged in SIT-12 and Ch-SIT-12. The other correspondingly colored sticks present the original sticks and its adapted version (sticks 13–16) for the Chinese SIT-12. Mean values (± standard error of the mean) of correct answer rates for every single Sniffin' Stick for all German (G, N = 100) and Chinese (Ch; N = 70) healthy subjects.

subjects as well as healthy controls and patients were compared using the Mann-Whitney-U-Test. The relationship of the variables age, education, smoking habits, disease severity and disease duration with Sniffin Sticks performance were investigated using the Spearman's correlation coefficient. Receiver Operating Characteristic Curves (ROC-curves) were calculated for the different odorant sets in both patient groups in comparison to age matched controls. Given the exploratory nature of the study, the results should not be interpreted as confirmatory. No adjustment for multiple tests was made. A p-value \ 0.05 was considered statistically significant.

## Results

### Demographic data

Healthy subjects as well as patients did not differ significantly according age or smoking habits. The German age matched controls showed a significantly longer education period in comparison to controls (p = 0.02), there was no such significant difference in Chinese patients and controls. There were more male than female patients but more female than male age matched controls; yet neither in patients nor in controls there was a significant difference in gender distribution between the Chinese and German cohorts. Also there was no significant difference in odor identification rates according to gender in all controls. Severity of motor symptoms (UPDRS III) did not differ significantly between both patient groups but disease duration did (p<0.0001) with a longer mean disease duration in German patients. A minor fraction of Chinese and German patients and controls were smokers. However, there was no significant difference in odor identification performance between smokers and non-smokers in any of the patient- and control-groups studied. Demographic Data are summarized in Table 1.

## Odor identification: Healthy probands

**Distribution of identification errors for single Sniffin' Sticks.** The rate of correct answers for each Sniffin' Stick (± standard error of the mean (SEM)) is depicted in Fig 1. For the original SIT-12 in the German control population all sticks were recognized by more than 75% except for Item 10 (test odor 'pineapple') which was recognized by 74% of controls.

For the Chinese group of healthy probands (cHP) sticks 2 (shoeleather, 47%), 3 (cinnamon, 67%), 6 (lemon, 72%) and 9 (cloves, 48%) of the original SIT-12 were recognized by less than 75% of probands. With the regionally adapted Chinese version and the change of odors for the sticks 3, 7, 8 and 9 only 2 sticks (2, 6) remained beneath 75% identification rate. Note that the remaining difference to 75% with sticks 2 and 6 was well within SEM borders.

**Comparison of German and Chinese healthy probands.** The whole group of Chinese and German HP (n = 170) showed a median score of correct answers of 11.0 (1.0) for SIT-12 and 11.0 (2.0) Ch-SIT-12. The difference was statistically significant (p = 0.02).

For the Chinese HP the median difference between the correct identifications of the SIT-12 (10.0 (2.0)) and Ch-SIT-12 (11.0 (1.25) was highly significant (p<0.0009). Instead the German HP showed a median of 11.0 (2.0) correct answers for SIT-12 and 11.0 (2.0) correct answers for Ch-SIT-12 (p = 0.8).

For the subgroups of Chinese and German young HP (yHP) the results of SIT-12 and Ch-SIT-12 showed the same pattern as for the HP groups. SIT-12 showed a median of 10.0 (2.0) correct answers for the Chinese yHP vs. 11.0 (2.0) for the German yHP. For Ch-ST-12 the median was 11.0 (2.0) for Chinese yHP vs 11.0 (3.0) for the German yHP. The differences in correct answers between the original and localized test were highly statistically significant for the Chinese yHP (p = 0.005) but not for the German yHP (p = 0.7).

In summary the Chinese HP and yHP showed significantly better performance in the adapted Ch-SIT-12 than in the SIT-12. Interestingly the German HP and yHP performed equally good in SIT-12 and Ch-SIT-12.

When comparing the performances of the intended test sets for each group of HP (SIT-12 for the German HP and Ch-SIT-12 for the Chinese HP) the difference in performance was much lower with no significant difference with a median of 11.0 (2.0) for German HP and 11.0 (1.25) for Chinese HP (p = 0.06).

The numbers of correct answers for each group are shown in Table 1 and are depicted in Fig 2.

## Odor identification: Comparison of patients and controls

**SIT-12.** For the German group there was a highly significant difference (p<0.0001) in odor identification abilities between patients (5.0 (4.5)) and controls (11.0 (1.25)). The same was true for the Chinese group (p<0.0001; patients: 7.0 (3.0); controls: 9.0 (3.0)). The Chinese patients performed better than the German patients, but German patients show a much longer disease duration than the Chinese with a median of 24.0 (38.0) vs. 96.0 (143.0) months in the German group.

**Ch-SIT-12.** The Ch-SIT-12 showed the same pattern of results as SIT-12 with significant differences between patients and controls for both the German (p<0.001, patients: 6.0 (4.5); controls: 11.0 (2.25) and the Chinese cohorts (p<0.001, patients 7.0 (4.0); controls: 10.0 (1.0)).

**Comparison of SIT-12 and Ch-SIT-12.** For both SIT-12 and Ch-SIT-12 Chinese PD performed better by approximately 1 item than German PD and both patient groups performed better by approximately 1 score in the Ch-SIT-12 in comparison to the SIT-12. Note again that the Chinese patient group has a significantly shorter mean disease duration than the German patient group.

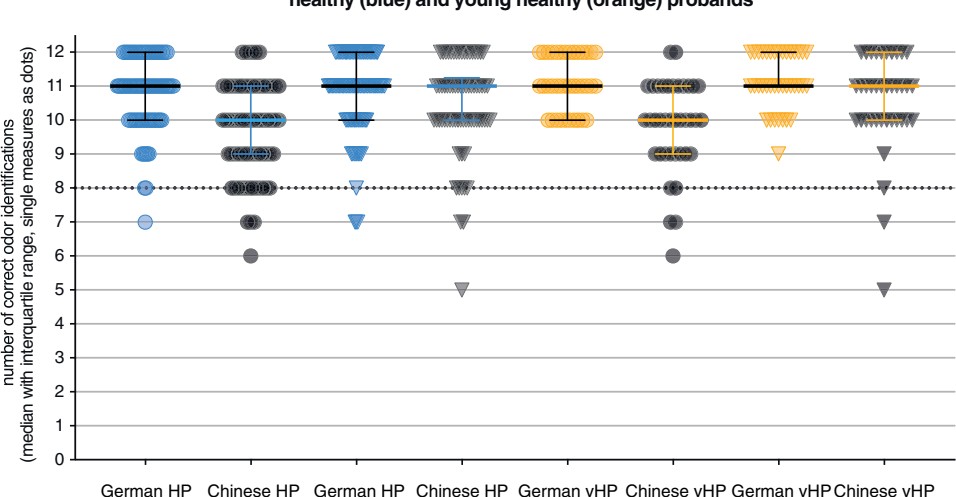

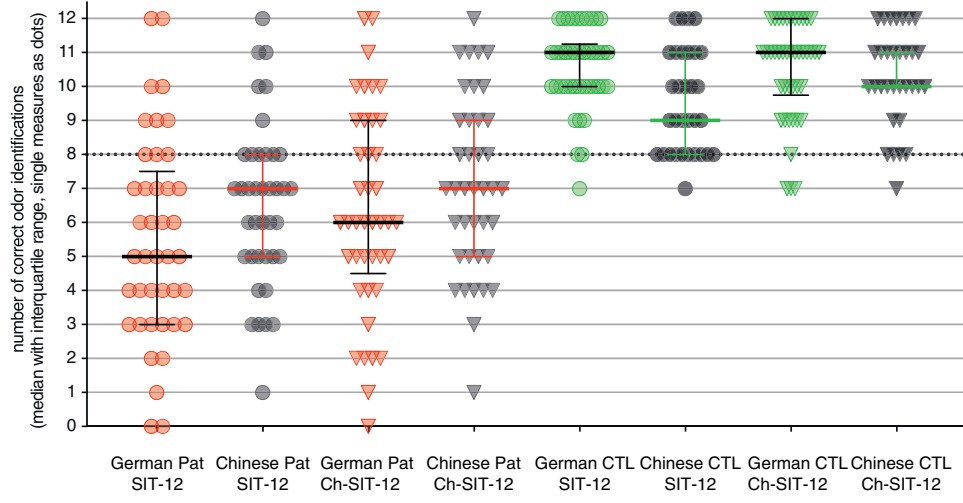

**Fig 2. Numbers of correct odor identifications as median with interquantile range and single measures as dots for all tested cohorts and subgroups as well as both tested Sniffin' Stick sets (SIT-12 = original German version; Ch-SIT-12 = adapted Chinese version).** Blue = all healthy subjects (HP); Orange = subgrop of young (20–40 years old) healthy subjects (yHP); Red = all patients (Pat), Green = age matched subgroup of healthy subjects as control group for patients (CTL).

When comparing the performance between the original SIT-12 and the Ch-SIT-12 the German controls as well as the German young healthy subjects performed nearly equally good.

For all patients and controls measurements the median scores and interquartile ranges are shown in Table 1. The median scores as well as every single patient and control measure are depicted in Fig 2.

## Sensitivity and specificity of SIT-12 and Ch-SIT-12

With a cutoff of < 7.5 for correct answers out of 12 sticks, the original SIT-12 test discriminated between German PD and HC with a sensitivity of 75% and a specificity of 98%. The

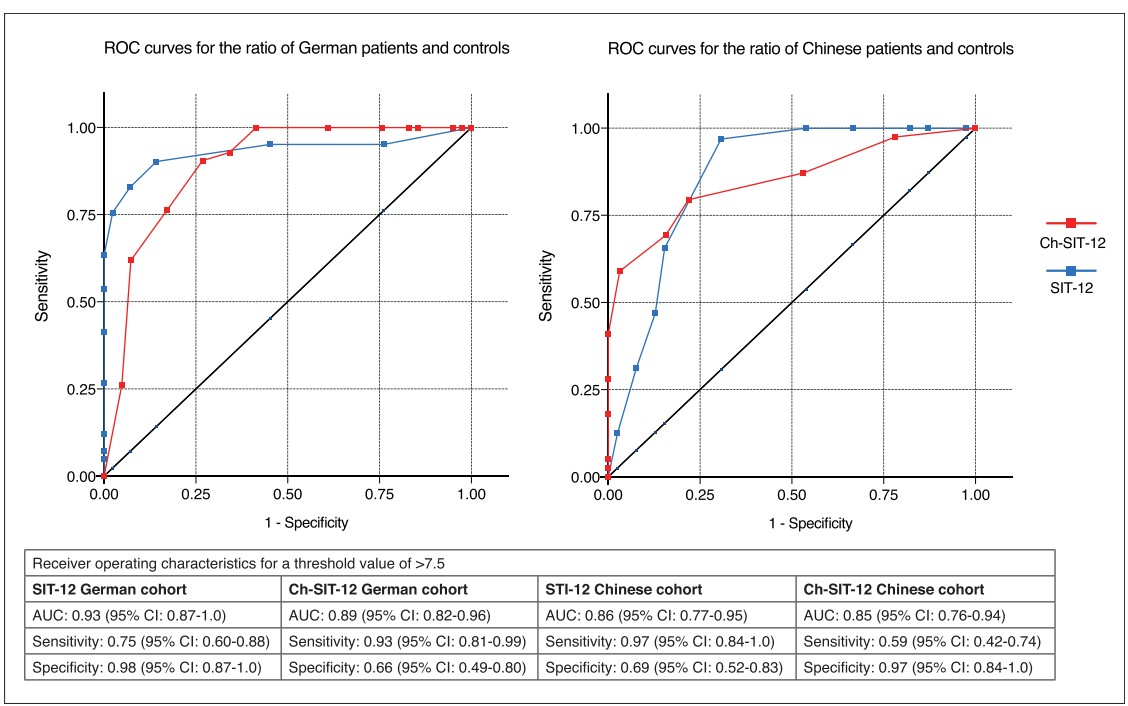

| Receiver operating characteristics for a threshold value of >7.5 | | | |
|---|---|---|---|
| **SIT-12 German cohort** | **Ch-SIT-12 German cohort** | **STI-12 Chinese cohort** | **Ch-SIT-12 Chinese cohort** |
| AUC: 0.93 (95% CI: 0.87-1.0) | AUC: 0.89 (95% CI: 0.82-0.96) | AUC: 0.86 (95% CI: 0.77-0.95) | AUC: 0.85 (95% CI: 0.76-0.94) |
| Sensitivity: 0.75 (95% CI: 0.60-0.88) | Sensitivity: 0.93 (95% CI: 0.81-0.99) | Sensitivity: 0.97 (95% CI: 0.84-1.0) | Sensitivity: 0.59 (95% CI: 0.42-0.74) |
| Specificity: 0.98 (95% CI: 0.87-1.0) | Specificity: 0.66 (95% CI: 0.49-0.80) | Specificity: 0.69 (95% CI: 0.52-0.83) | Specificity: 0.97 (95% CI: 0.84-1.0) |

**Fig 3. Receiver operating characteristics (ROC-curves) for a threshold value of > 7.5 for the comparison of German patients and controls (left) and Chinese patients and controls (right) for the original SIT-12 (blue) and the Chinese adapted version Ch-SIT-12 (red).**

ROC curve showed an area under curve of 0.93. With the regionalized Ch-SIT-12 test the Chinese cohort of PD and HC showed a sensitivity of 59% and specificity of 97% with an area under curve of 0.85. Note that the cross-testing (regionalized Chinese SIT with German cohort and original SIT-12 with Chinese cohort) shows a higher sensitivity with lower specificity both for Germany and China. ROC-curves and according numbers are depicted in Fig 3.

## Possible factors influencing odor identification

Correlation analysis showed no significant relationship for younger age (r = -0.14, p = 0.08) but a significant relation to longer education (r = 0.16, p = 0.03) with a better performance over all 16 Sniffin' Sticks used in the group of 170 healthy probands. For the patient group of 80 PD there was no significant relation to age, education and disease duration but there was a significant relation with higher (worse) UPDRS III scores (r = -0.26, p = 0.03) and worse performance in the Sniffin Sticks.

## Discussion

Odor identification is a simple yet useful tool that helps to discriminate PD from other causes of parkinsonism [2]. However, smells are subject to cultural differences, so that test procedures have to be adapted to the regional circumstances and habits where they are intended for use [25]. In this study we investigated a regionalized set of odors for the SIT-12 test for a Chinese population. We found that the modified Ch-SIT-12 showed the best specificity but fairly low sensitivity to discriminate the Chinese PD patients from the age matched controls cohort. The original German SIT-12 test proved to be the best test-constellation to discriminate the

German cohort of patients and controls and showed a high sensitivity at the cost of a low specificity in the Chinese cohort.

Krismer et al. [26] found that the identification test had a similar area under curve compared with the sum score of the whole Sniffin' Sticks test battery, moreover the identification test was superior to any of the other subtests that are threshold and discrimination. In this study we examined one of the most commonly used European odor screening identification tests, the SIT-12 for its application in Germany and a northern Chinese (Beijing) environment in the original German version and in a regionally adapted Chinese version (Ch-SIT-12). The special feature of this study was that the close cooperation of the working groups in Beijing and Ulm made it possible to conduct and compare the examinations at the same time on adapted German and Chinese collectives.

It is of note that the Chinese and German patient cohorts showed a significantly different disease duration with a much shorter median duration for the Chinese patients. However, Nielsen et al [27] comment, that the majority of studies that investigated olfactory dysfunction as a possible progression marker did not find disease duration to be a significant independent predictor of the olfactory score.

In comparison to one of the most recent studies of the unmodified SIT-12 test in China by Huang et al. [28] we could not replicate the high specificity of 81.5% but we elicited the same tendency to a relatively lower specificity and a high sensitivity at the commonly used cut off value of > 7.5 correct answers. The same tendency can be seen in SIT-12 testing in Spain where López Hernández et al showed 70% sensitivity and 83% specificity [29].

As depicted in the review of olfactory testing procedures in PD by Nielsen et al. [27] the sensitivity and specificity values vary considerably. Also Krismer et al [26] could show that manipulating the threshold for cut-off values for the identification test resulted in either excellent sensitivity or high specificity in discriminating PD patients from HC. Krismer et al [26] point out that values of high specificity are useful to confirm a diagnosis. Regardsing this, it is noteworthy that both in German and Chinese probands the respective "non-regionalized" versions of the SIT-12 the ROCs showed a different pattern with high sensitivity scores at the expense of specificity. This is a non-favorable characteristic for a test that is intended to be used as a differential-diagnostic tool for a neurodegenerative disease.

The SIT-12 test demands >75% correct answers for each odorant in a normosmic population [14,24]. Except for Stick 10 (pineapple) with 74% mean correct answers this was accomplished for the German HPs. For the Chinese cohort of HPs, Sniffin' Stick 2 (shoeleather) showed an unexpectedly low recognition rate, especially since an exchange of this stick against a regionalized version was not provided. For those odors for which an exchange was anticipated and implemented in Ch-SIT-12, a mixed picture emerges. Cloves that showed a low recognition rate of under 50% in den Chinese HP was replaced by garlic with a now high rate of 97%. Yet liquorice and coffee that were replaced by soy sauce and chocolate odors were already recognized in the original version so well that an exchange would not have been necessary. To sum up the single Sniffin' Stick characteristics, the adapted Ch-SIT-12 test worked as intended with correct answer rates close to (within SEM borders) or well above 75%, except for stick 2.

Most interestingly the German cohort of healthy subjects performed excellent also in the Ch-SIT-12 test with no statistical difference to the original test (Fig 2). However, the Chinese healthy probands showed a significantly worse performance in the original SIT-12 in comparison to Ch-SIT-12. By contrast, the German HPs showed a very good detection rate also for all of the Chinese replacement odors. The reasons can only be speculated, but it is well known that perception relies on previously acquired knowledge [30]. Therefore, it seems plausible, that the great popularity and widespread availability of Chinese food in Germany could be a major factor. Related to this was the assumption that young adults may do even better than

older people. However, this could not be confirmed in the investigation. Although the known age-related worsening of the olfactory function [17] was found to be significant both in the Chinese and German HPs, there was no significant difference between the characteristic Chinese and German odors in the older controls in comparison to the younger ones.

The study has several shortcomings that need to be addressed. An ENT examination was not carried out in order to exclude other diseases with olfactory dysfunction. Yet it was asked for known diseases and for subjective normal smelling ability before inclusion of the probands. In addition, the study design with the coupled examination of the SIT-12 and Ch-SIT-12 testsets is unusual. However, only through this procedure it was possible to avoid a re-test effect of the 8 unchanged odors within one subject.

In summary, we found that the modified SIT-12 test offers a quick way to screen for hyposmia in PD patients in a Chinese population. In retrospective the exchange of the odors liquorice and coffee would not have been necessary, as these odors showed equally high recognition rates in Chinese and German probands. However, also the replacement odors performed well. According to the one remaining odor (shoeleather) that performed much lower than the demanded 75% recognition rate, it will be necessary to investigate in a subsequent study if the adaption of this odor will be able to further improve test performance.

## Supporting information

**S1 Table. Minimal data set.** Complete data set for Sniffin' Sticks results, demographic data and clinical scores for Chinese and German patients, controls and healthy probands. (XLSX)

## Author Contributions

**Conceptualization:** Elmar H. Pinkhardt, Albert C. Ludolph, Yining Huang, Zhaoxia Wang.

**Data curation:** Elmar H. Pinkhardt, Huijing Liu, Adrian Pachollek, Martin S. Kunz.

**Formal analysis:** Elmar H. Pinkhardt.

**Funding acquisition:** Elmar H. Pinkhardt, Albert C. Ludolph, Yining Huang, Haibo Chen, G. Bernhard Landwehrmeyer.

**Investigation:** Elmar H. Pinkhardt, Huijing Liu, Di Ma, Jing Chen, Adrian Pachollek, Martin S. Kunz, Zhaoxia Wang, Wen Su.

**Methodology:** Elmar H. Pinkhardt.

**Project administration:** Elmar H. Pinkhardt.

**Supervision:** Elmar H. Pinkhardt.

**Writing – original draft:** Elmar H. Pinkhardt.

**Writing – review & editing:** Huijing Liu, Di Ma, Jing Chen, Adrian Pachollek, Martin S. Kunz, Jan Kassubek, Albert C. Ludolph, Yining Huang, Haibo Chen, G. Bernhard Landwehrmeyer, Zhaoxia Wang, Wen Su.

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
