## [Decision Letter · Decision Letter 0]

24 Jul 2019

PONE-D-19-15889

Olfactory Screening of Parkinson’s Disease patients and healthy subjects in China and Germany: A study of cross-cultural adaptation of the Sniffin’ Sticks 12-identification test

PLOS ONE

Dear Prof. Dr. Pinkhardt,

Thank you for submitting your manuscript to PLOS ONE. After careful consideration, we feel that it has merit but does not fully meet PLOS ONE’s publication criteria as it currently stands. Therefore, we invite you to submit a revised version of the manuscript that addresses the points raised during the review process.

We would appreciate receiving your revised manuscript by Sep 07 2019 11:59PM. To enhance the reproducibility of your results, we recommend that if applicable you deposit your laboratory protocols in protocols.io, where a protocol can be assigned its own identifier (DOI) such that it can be cited independently in the future. For instructions see: http://journals.plos.org/plosone/s/submission-guidelines#loc-laboratory-protocols

We look forward to receiving your revised manuscript.

Kind regards,

Jing A. Zhang, MD, PhD

Academic Editor

PLOS ONE

Journal Requirements:

3. Please upload a copy of Figure 5, to which you refer in your text on page 13. If the figure is no longer to be included as part of the submission please remove all reference to it within the text.

Reviewers' comments:

Reviewer's Responses to Questions

**Comments to the Author**

1. Is the manuscript technically sound, and do the data support the conclusions?

Reviewer #1: Yes

Reviewer #2: Partly

2. Has the statistical analysis been performed appropriately and rigorously? 

Reviewer #1: Yes

Reviewer #2: No

3. Have the authors made all data underlying the findings in their manuscript fully available?

Reviewer #1: Yes

Reviewer #2: Yes

4. Is the manuscript presented in an intelligible fashion and written in standard English?

Reviewer #1: Yes

Reviewer #2: Yes

5. Review Comments to the Author

Reviewer #1: Review

PLPS one manuscrpit PONE D 19 15889

By inkhardt E et al.

This manuscript reports the first study to directly compare olfactory function in a North Chinese and a German cohort of healthy subjects and Parkinson patients. For testing olfactory function the SIT12 (Sniffin Sticks) test and a modified CH-SIT12 test – the later adapted to the Chinese culture – are employed.

The study is explanatory and therefore does not fulfill the strict criteria of a test-retest development with all the statistical ramifications. By testing all 16 odors in one test procdur, the need for adjusting the analysis for multiple testing was avoided.

There are a few comments:

Line 86 is not clear – cited: for the patient groups we likewise hypothesize …… please change

Line 119: was the UPDR III rating performed in On or Off – I assume the PD patients were under treatment.

Line 167: Is there evidence that olfactory function deteriorates during the progression of PD or is the hyposmia stable over time ?

Please quote respective references.

Table 1

20 % of German PD patients were smokers. Did they perform differently ?

Were the Chinese PD patients de novo patients ? They had a disease duration of about 3 years.

Was the Sniffin Test performed by asking the person to sniff once or twice ? – there is a difference in the results if you sniff once or twice

Line 191-199: For a naive reader this paragraph is slightly confusing. Please try to present the data in a simpler way.

Discussion:

Line 312: The most likely explanation of the results is the fact that German subjets are widely exposed to Chinese Cuisine.

It is not surprising that liquorice and coffee are well recognized by the Chinese Healthy controls and PD patients. This is part of their day to day diet . Thus it is not clear why these two odors where exchanged. This should be explained in the methods.

Conclusion: it may be changed. As in the strict sense the CH-SIT12 would only need an exchange of 2 odors and a replacement for the oder shoeleather.

Criteria for publications:

Original

Trans-national

Nicely carried out

Minor modifications requested.

Reviewer #2: The authors present an interesting study on the cross-cultural adapted olfactory screening test (Sniffin’ Sticks 12-identification test) for the differentiation between Parkinson’s disease and healthy subject. The purpose of the work is meaningful as we indeed sense the impact of cultural difference on the accuracy of olfactory screening tests in clinical practice. Yet, I have the following concerns that need to be addressed.

1. The statistics are incorrect. The student t test can be used in data with normal distribution. The author did not indicate the feature of the data. Judging from table 1, non-normal distribution is very likely for disease duration, smoking years, UPDRS III scores in this cohort. If that’s the case, the data should be firstly displayed as median value and interquartile range, then analyzed using Kruskal-Wallis test. Likewise, Pearson correlation is used in normally distributed data as well. The authors need to indicate the distribution of the data, and rectify the analytic approach and the result accordingly.

2. The conclusion that the adapted Chinese version screening test (Ch-SIT-12) is more useful than the original SIT-12 can’t be reached according to the result. The ROC curve shows that the non-adapted version has higher AUC than the adapted version. Besides, as the disease duration and gender differ between Chinese and German patients, the superiority may not be suitable to analyze in the first place.

3. Can the authors indicate who conducted the olfactory test for the subjects? Is it the same group performing the olfactory test for these subjects? Have they been trained in the same way? Is there a common protocol for both German and Chinese practitioners? Is there any blindness used? Is there inter-rater variability tested? I’m concerned as Chinese subjects performed better on both the non-adapted (SIT-12) and the adapted olfactory test (ch-SIT-12). Besides, the curve in ROC figure is smoother in German subjects while a small bump can be seen in the Chinese subjects, indicating larger variability in Chinese subject. According to Table 1, larger SD is shown. Can the variability come from different raters?

4. Have the abstract finished? There is an unfinished sentence in the result parts. There are grammar errors in the abstract. And the expression used in the abstract need to be more simplified and concise.

5. I suggest a brief summary of the main finding and implication of this study in the first paragraph in the discussion part.

6. Several grammar errors are present across the article. And th

6. PLOS authors have the option to publish the peer review history of their article (what does this mean?). If published, this will include your full peer review and any attached files.

Reviewer #1: No

Reviewer #2: No

---

## [Author Response · Author response to Decision Letter 0]

24 Aug 2019

Revised manuscript PONE-D-19-15889: Olfactory Screening of Parkinson’s Disease patients and healthy subjects in China and Germany: A study of cross-cultural adaptation of the Sniffin’ Sticks 12-identification test

In the following, we reply to the reviewers’ comments on a point-to-point basis.

Reviewer #1: 

This manuscript reports the first study to directly compare olfactory function in a North Chinese and a German cohort of healthy subjects and Parkinson patients. For testing olfactory function the SIT12 (Sniffin Sticks) test and a modified CH-SIT12 test – the later adapted to the Chinese culture – are employed.

The study is explanatory and therefore does not fulfill the strict criteria of a test-retest development with all the statistical ramifications. By testing all 16 odors in one test procdur, the need for adjusting the analysis for multiple testing was avoided.

There are a few comments:

Line 86 is not clear – cited: for the patient groups we likewise hypothesize …… please change

The sentence was changed to: For the patient groups we hypothesized that both the Chinese as well as the German PD patients would perform worse in the non-adapted version of the test in comparison to the adapted version.

Line 119: was the UPDR III rating performed in On or Off – I assume the PD patients were under treatment.

The reviewer’s assumption is correct; the patients were rated in the On state. The following sentence was added to the manuscript: UPDRS-III was performed under ongoing treatment in the On state.

Line 167: Is there evidence that olfactory function deteriorates during the progression of PD or is the hyposmia stable over time ?

Please quote respective references.

The reviewer point to an important question as we found a significant difference in disease duration between the Chinese and the German patient group. However, the corresponding literature on this topic does not show a clear picture. Also in our patient cohorts we did not find a significant correlation between disease duration and sniffin’ sticks performance. 

To address the topic, the following paragraph was added to the discussion section of the manuscript: It is of note that the Chinese and German patient cohorts showed a significantly different disease duration with a much shorter median duration for the Chinese patients. However, Nielsen et al (Nielsen et al., 2018) comment, that the majority of studies that investigated olfactory dysfunction as a possible progression marker did not find disease duration to be a significant independent predictor of the olfactory score.

Table 1

20 % of German PD patients were smokers. Did they perform differently?

There was no statistically significant difference in odor identification performance between smokers and non-smokers in any of the cohorts studied (German patients and controls, Chinese patients and controls). The corresponding results were added to the results section of the manuscript reading: “…A minor fraction of Chinese and German patients and controls were smokers. However, there was no significant difference in odor identification performance between smokers and non-smokers in any of the patient- and control-groups studied.”

Were the Chinese PD patients de novo patients ? They had a disease duration of about 3 years.

The Chinese patients were not de novo patients. 

Was the Sniffin Test performed by asking the person to sniff once or twice ? – there is a difference in the results if you sniff once or twice.

According to the suggested procedure by Hummel et al (Hummel T, Sekinger B, Wolf SR, Pauli E, Kobal G. “Sniffin” sticks’: olfactory performance assessed by the combined testing of odor identification, odor discrimination and olfactory threshold. Chem Senses 1997 Feb 1;22(1):39–52) for odor presentation the cap was removed by the experimenter for ~3 s and the pen's tip was placed ~2 cm in front of both nostrils. 

The corresponding sentence in the methods section has been changed as follows: . “… According to the suggested procedure by Hummel et al (Hummel et al., 1997) the odor sticks are successively placed about 2-3 cm in front of the nose of the subject for a period of about 3 seconds.

Line 191-199: For a naive reader this paragraph is slightly confusing. Please try to present the data in a simpler way.

The paragraph was rewritten and reads now: For the subgroups of Chinese and German young HP (yHP) the results of SIT-12 and Ch-SIT-12 showed the same pattern as for the HP groups. SIT-12 showed a median of 10.0 (2.0) correct answers for the Chinese yHP vs. 11.0 (2.0) for the German yHP . For Ch-ST-12 the median was 11.0 (2.0) for Chinese yHP vs 11.0 (3.0) for the German yHP. The differences in correct answers between the original and localized test were highly statistically significant for the Chinese yHP (p=0.005) but not for the German yHP (p=0.7).

In summary the Chinese HP and yHP showed significantly better performance in the adapted Ch-SIT-12 than in the SIT-12. Interestingly the German HP and yHP performed equally good in SIT-12 and Ch-SIT-12.

Discussion:

Line 312: The most likely explanation of the results is the fact that German subjets are widely exposed to Chinese Cuisine.

It is not surprising that liquorice and coffee are well recognized by the Chinese Healthy controls and PD patients. This is part of their day to day diet . Thus it is not clear why these two odors where exchanged. This should be explained in the methods.

The decision to change the above-mentioned odors was based on the familiarity ratings of odors as described by Shu et al (Shu C-H, Yuan B-C. Assessment of odor identification function in Asia using a modified “Sniffin’ Stick” odor identification test. Eur Arch Oto-Rhino-Laryngology. 2007 Dec 7;265(7):787–90) and the experience of the Chinese authors of this study.

This paragraph has been added to the methods part of the manuscript.

Conclusion: it may be changed. As in the strict sense the CH-SIT12 would only need an exchange of 2 odors and a replacement for the odor shoeleather.

According to the reviewer’s suggestion the conclusion has been rewritten and reads now: “…In summary, we found that the modified SIT-12 test offers a quick way to screen for hyposmia in PD patients in a Chinese population. In retrospective the exchange of the odors liquorice and coffee would not have been necessary, as these odors showed equally high recognition rates in Chinese and German probands. However, also the replacement odors performed well. According to the one remaining odor (shoeleather) that performed much lower than the demanded 75% recognition rate, it will be necessary to investigate in a subsequent study if the adaption of this odor will be able to further improve test performance.”

Reviewer #2: 

The authors present an interesting study on the cross-cultural adapted olfactory screening test (Sniffin’ Sticks 12-identification test) for the differentiation between Parkinson’s disease and healthy subject. The purpose of the work is meaningful as we indeed sense the impact of cultural difference on the accuracy of olfactory screening tests in clinical practice. Yet, I have the following concerns that need to be addressed.

1. The statistics are incorrect. The student t test can be used in data with normal distribution. The author did not indicate the feature of the data. Judging from table 1, non-normal distribution is very likely for disease duration, smoking years, UPDRS III scores in this cohort. If that’s the case, the data should be firstly displayed as median value and interquartile range, then analyzed using Kruskal-Wallis test. Likewise, Pearson correlation is used in normally distributed data as well. The authors need to indicate the distribution of the data, and rectify the analytic approach and the result accordingly.

According to the reviewer’s suggestion the statistics have been newly calculated. Now all data in the text and the table (except for age) are displayed as median and interquantile rage. Due to the non-normal distribution of the data, non-parametric tests were used. The Chinese and German subjects as well as healthy controls and patients were compared using the Mann-Whitney-U-Test. The relationship of the variables age, education, smoking habits, disease severity and disease duration with sniffin sticks performance were investigated using the Spearman’s correlation coefficient.

2. The conclusion that the adapted Chinese version screening test (Ch-SIT-12) is more useful than the original SIT-12 can’t be reached according to the result. The ROC curve shows that the non-adapted version has higher AUC than the adapted version. Besides, as the disease duration and gender differ between Chinese and German patients, the superiority may not be suitable to analyze in the first place.

According to the reviewer’s suggestion the conclusion has been rewritten and reads now as follows: “…In summary, we found that the modified SIT-12 test offers a quick way to screen for hyposmia in PD patients in a Chinese population. In retrospective the exchange of the odors liquorice and coffee would not have been necessary, as these odors showed equally high recognition rates in Chinese and German probands. However, also the replacement odors performed well. According to the one remaining odor (shoeleather) that performed much lower than the demanded 75% recognition rate, it will be necessary to investigate in a subsequent study if the adaption of this odor will be able to further improve test performance.”

3. Can the authors indicate who conducted the olfactory test for the subjects? Is it the same group performing the olfactory test for these subjects? Have they been trained in the same way? Is there a common protocol for both German and Chinese practitioners? Is there any blindness used? Is there inter-rater variability tested? I’m concerned as Chinese subjects performed better on both the non-adapted (SIT-12) and the adapted olfactory test (ch-SIT-12). Besides, the curve in ROC figure is smoother in German subjects while a small bump can be seen in the Chinese subjects, indicating larger variability in Chinese subject. According to Table 1, larger SD is shown. Can the variability come from different raters?

All raters have been trained in the same way. There was a detailed and common protocol for othe German and Chinese practitioners. The test procedure was the one, described by Hummels et al (Hummel T, et al. “Sniffin” sticks’: olfactory performance assessed by the combined testing of odor identification, odor discrimination and olfactory threshold. Chem Senses. 1997 Feb 1;22(1):39–52.) which is the standard procedure for the test. 

Due to the nature of the test procedure and the task of the raters there is no need for a blinded rater and/or inter-rater variability. All raters strictly followed the standardized procedures i.e. presenting the sticks to both nostrils for 3 seconds each in a distance of 2-3 centimeters and document the answer of the proband

.

4. Have the abstract finished? There is an unfinished sentence in the result parts. There are grammar errors in the abstract. And the expression used in the abstract need to be more simplified and concise.

The unfinished sentence has been removed. The abstract has been revised according to the reviewer’s suggestion

5. I suggest a brief summary of the main finding and implication of this study in the first paragraph in the discussion part.

Following the reviewer’s suggestions the first paragraph was rewritten and reads now: “Odor identification is a simple yet useful tool that helps to discriminate PD from other causes of parkinsonism (2). However, smells are subject to cultural differences, so that test procedures have to be adapted to the regional circumstances and habits where they are intended for use (25). In this study we investigated a regionalized set of odors for the SIT-12 test for a Chinese population. We found that the modified Ch-SIT-12 showed the best specificity but fairly low sensitivity to discriminate the Chinese PD patients from the age matched controls cohort. The original German SIT-12 test proved to be the best test-constellation to discriminate the German cohort of patients and controls and showed a high sensitivity at the cost of a low specificity in the Chinese cohort.”

6. Several grammar errors are present across the article. 

The article has been carefully revised for grammar errors.

---

## [Decision Letter · Decision Letter 1]

11 Oct 2019

Olfactory Screening of Parkinson’s Disease patients and healthy subjects in China and Germany: A study of cross-cultural adaptation of the Sniffin’ Sticks 12-identification test

PONE-D-19-15889R1

Dear Dr. Pinkhardt,

We are pleased to inform you that your manuscript has been judged scientifically suitable for publication and will be formally accepted for publication once it complies with all outstanding technical requirements.

With kind regards,

Jing A. Zhang, MD, PhD

Academic Editor

PLOS ONE

Additional Editor Comments (optional):

Reviewers' comments:

Reviewer's Responses to Questions

**Comments to the Author**

1. If the authors have adequately addressed your comments raised in a previous round of review and you feel that this manuscript is now acceptable for publication, you may indicate that here to bypass the “Comments to the Author” section, enter your conflict of interest statement in the “Confidential to Editor” section, and submit your "Accept" recommendation.

Reviewer #1: (No Response)

2. Is the manuscript technically sound, and do the data support the conclusions?

Reviewer #1: Yes

3. Has the statistical analysis been performed appropriately and rigorously? 

Reviewer #1: Yes

4. Have the authors made all data underlying the findings in their manuscript fully available?

Reviewer #1: Yes

5. Is the manuscript presented in an intelligible fashion and written in standard English?

Reviewer #1: Yes

6. Review Comments to the Author

Reviewer #1: No comments.

7. PLOS authors have the option to publish the peer review history of their article (what does this mean?). If published, this will include your full peer review and any attached files.

Reviewer #1: No

---

## [Editor Report · Acceptance letter]

29 Oct 2019

PONE-D-19-15889R1 

Olfactory Screening of Parkinson’s Disease patients and healthy subjects in China and Germany: A study of cross-cultural adaptation of the Sniffin’ Sticks 12-identification test 

Dear Dr. Pinkhardt:

I am pleased to inform you that your manuscript has been deemed suitable for publication in PLOS ONE. Congratulations! Your manuscript is now with our production department. 

With kind regards,

on behalf of

Dr. Jing A. Zhang 

Academic Editor

PLOS ONE